# Characterization of a Novel POx-Based Adhesive Powder for Obliterating Dead Spaces After Surgery

**DOI:** 10.3390/bioengineering12101011

**Published:** 2025-09-23

**Authors:** Steven E. M. Poos, Roger M. L. M. Lomme, Edwin A. Roozen, Johan C. M. E. Bender, Harry van Goor, Richard P. G. Ten Broek

**Affiliations:** 1Department of Surgery (Route 618), Radboud University Medical Center, P.O. Box 9101, 6500 HB Nijmegen, The Netherlands; 2GATT Technologies BV, Transistorweg 5, 6534 AT Nijmegen, The Netherlands

**Keywords:** dead space, seroma, POx-based powder, tissue adhesive, Poly(oxazolines), general surgery

## Abstract

Surgical dead spaces are challenging to handle with current preventive methods. Tissue adhesives show promise in obliterating ‘dead spaces’, but the drawbacks of currently available adhesives prevent them from being used for dead space elimination. An adhesive powder based on N-Hydroxysuccinimide-poly(2-oxazoline), NHS-POx, combines robust adhesive strength in moist environments with favorable biocompatibility and biodegradability, which makes this an interesting candidate for eliminating spaces that remain between tissues after surgery. The current study evaluates the swelling, crosslinking speed, and degradation properties of this novel tissue adhesive. These results were then used to design multiple adhesive variants differing in pH, surfactant addition, and particle size, which were subsequently examined based on their wetting rates, adhesive strength, and durability. The powder displayed minimal swelling and rapid crosslinking properties, by which the latter could be increased by a basic buffer or surfactant addition and reduced by increasing particle size. The wetting rate of the powder increased when a surfactant (Pluronic F68) was added to the mix. The adhesive strength, as measured by tensile and shear strength measurements of different prototypes of the adhesive powder, was significantly better than that of a commercially available fibrin glue. The addition of both buffer and Pluronic F68 led to a breakdown of adhesive force after 14 days of incubation, while the prototype containing neither buffer nor Pluronic F68 still had measurable adhesive force after 14 days of incubation. The current study results display several characteristics of the NHS-POx-based tissue adhesive that are favorable for tissue approximation, preventing the occurrence of dead spaces. The most effective and usable adhesive prototype will be identified in further ex vivo and in vivo animal model studies.

## 1. Introduction

Subcutaneous spaces can persist after skin closure following surgical resection or tissue flap dissection, which may become filled with blood or exudate. This fluid accumulation creates an environment conducive to bacterial infections, which can compromise recovery and delay hospital discharge [1,2,3]. Surgeons most commonly use closed wound drainage systems to prevent fluid accumulation in pockets between tissue layers, so-called dead spaces, even though conflicting evidence exists that drains prevent seroma formation [4,5]. Moreover, drains can facilitate exogenous bacterial invasion [6,7] and often lead to significant discomfort for patients, negatively impacting their quality of life [8]. In recent years, surgeons have used more complex techniques such as quilting to obliterate dead space. While these methods have shown some success in limiting dead space and seroma formation, their applicability is constrained by the necessity for specialized skills and the potential to significantly prolong operative time [9,10].

Tissue adhesives are a potential alternative for obliterating dead spaces, but new adhesives are needed to fulfill this potential [11,12]. Tissue adhesives are faster, less painful, and less traumatic to the tissue, while their application requires fewer specialized skills than suturing techniques or stapling devices. Clinical studies have pointed out that the use of tissue adhesives decreases pain and procedure time both in the emergency room and in the inpatient setting [13,14]. However, current adhesives have drawbacks that limit their broad applicability in clinical practice. Biologically produced adhesives have been used ever since the first fibrin glues were introduced in the 1990s [15]. These adhesives, primarily composed of fibrinogen and thrombin, exhibit excellent biocompatibility and cohesion. However, they often lack sufficient adhesive strength to tissue [16]. Other biological adhesives based on gelatin, albumin, or glutaraldehyde demonstrate stronger adhesion properties but are associated with cytotoxicity concerns [17,18,19]. Researchers came up with ways to address the biocompatibility issues of these adhesives, for example, by using double-bonding techniques or creating covalently cross-linked matrices [20,21,22]. While these strategies have improved biocompatibility, they have often resulted in decreased adhesive effectiveness and increased settling time [23]. Adhesives with polyethylene glycol (PEG) backbone have shown promising performance due to their rapid and robust adhesive strength and sufficient biocompatibility, but mechanical issues related to extensive swelling have limited their application for internal wound closure [24,25,26]. A next-generation tissue adhesive should be biologically compatible and non-toxic, should form a robust adhesion to tissue, should be mechanically similar to neighboring tissues, should be fast and easy to use, and should have biodegradation characteristics that minimally hinder wound healing processes.

The synthetic polymer N-Hydroxysuccinimide-poly(2-oxazoline), NHS-POx, has the potential to combine robust adhesive strength in moist environments with excellent biocompatibility and biodegradability. NHS-POx is a copolymer of 2-n-propyl-2-oxazoline and 2-methoxycarbonylethyl-2-oxazoline that contains NHS-ester as side-chain moiety [27]. Polyoxazoline polymers offer the flexibility to incorporate side chains on a per-monomer basis, rendering them highly tunable for a wide range of specific applications [28]. The NHS-ester has been used in various synthetic adhesives because of its low toxicity and high reactivity to form amide groups with amines that are abundantly present in various tissues [12,29,30]. NHS-POx is able to form a stronger polymer network compared to NHS-PEG due to its higher density of NHS-ester groups, while being completely biocompatible upon subcutaneous application [31]. NHS-POx has been used together with nucleophilic-activated polyoxazoline (NU-POx) in a gelatin patch, where it demonstrated its hemostatic and aerostatic abilities [32,33,34,35]. Also, a combination of NHS-POx and POx-Alendronate combined in a gelatin carrier has been previously examined as a potential membrane barrier for dental defects, where it displayed a strong adhesion to bone tissue [36,37,38].

We have developed an adhesive powder based on NHS-POx intended to obliterate dead space and prevent hematoma and seroma after soft tissue and tissue flap surgery. When applied, the powder is sprayed on a dried wound site to form a thin layer. During wound closure, moisture present enables covalent amide bonds to form between NHS-ester sites on NHS-POx and amine sites present in the powder and in tissue. These bonds help bridge the defect, effectively closing the dead space. The powder consists of ingredients that add their own functionality to the NHS-POx component. NU-POx is added to increase the covalent crosslinking density within the polymer matrix. Crosslinked gelatin is used as the water-insoluble backbone for the NHS-POx/NU-POx polymer matrix while also playing a part in the internal strength of the polymer matrix through the abundance of amine groups present in peptides and proteins of the gelatin. Trehalose, a well-known expedient in drug- and medical device development, is added as a dispersant to improve the powder’s sprayability as well as to ensure an evenly distributed crosslinking reaction to take place upon contact with an aqueous solvent [39,40]. It is known that the powder’s pH, hydrophilicity, and particle size play crucial roles in the eventual water uptake and adhesive capabilities of this novel adhesive powder [41]. Increasing the powder’s pH by adding a buffer could improve NHS-ester amide binding, as the optimum pH for NHS-ester reactivity is at pH 8.5 [33], although this pH is also the optimum for hydrolyzing NHS-ester groups [42]. The hydrophilicity could be improved by adding a surfactant, which would increase the solubility of the powder in aqueous solvent, possibly increasing wetting capability and crosslinking speed as well as adhesive strength. Differences in particle size impact the powder’s surface area directly exposed to solvent, which is expected to lead to differences in sprayability, water uptake, crosslinking speed, and potentially adhesive strength.

This study consists of a series of experiments to formulate POx-based adhesive powders that have the potential to be applied in surgery for obliterating dead spaces. First, the powder, its ingredients, and intermediate products were chemically characterized and altered to determine the parameters that influence the powder’s performance and usability. With the information gained from these characterization experiments, we developed fourteen different powder prototypes with varying pH, hydrophilicity, and particle size, and compared them with a commercially available fibrin glue on key performance indicators, e.g., water uptake, adhesive strength, and durability in in vitro experiments.

## 2. Materials and Methods

### 2.1. Study Outline

This study is divided into two sections. First, the wetting uptake of individual ingredients was assessed, and the base adhesive powder was characterized by swelling rate, crosslinking speed, and in vitro degradation. Using these findings, multiple powder prototypes varying in particle size, buffer-pH, and/or surfactant content were developed and evaluated for water uptake and adhesive strength capabilities (Figure 1).

### 2.2. Materials

The polymers NHS-POx, POx-OH-COOH, and NU-POx were produced by GATT Technologies (Nijmegen, The Netherlands), according to protocols described in the literature [32]. Crosslinked gelatin powder was obtained from Mascia Brunelli S.p.A (Milan, Italy). Trehalose dihydrate, Borax(di-sodium tetraborate), sodium carbonate, and Pluronic F-68 (BASF Pharma, Ludwigshafen, Germany) were purchased from VWR (Radnor, PA, USA). Fibrin glue, TISSEEL^®^, was obtained from Baxter International (Deerfield, IL, USA).

### 2.3. Preparation of Adhesive Powder

The formulation and ratio of ingredients were based on preliminary experiments described in patent WO2022148725 [41]. In the first step of the production process, trehalose, crosslinked gelatin powder, and NU-POx were mixed with demineralized water using a Turrax mixer (IKA, Staufen, Germany). The temperature of the demineralized water was 40 °C to prevent crosslinked gelatin from hydrolyzing during this process; at the same time, it ensured that gelatin would organize into a random coil formation, making it possible for NU-POx and trehalose to mix well with gelatin macromers. Once the solution cooled, gelatin monomers coacervated into a three-dimensional network with the trehalose and NU-POx impregnated therein [43]. The mix was freeze-dried and ground into a powder, which will be referred to as NU-phase, using a coffee grinder (Retsch, Haan, Germany) and pestle and mortar (VWR, Radnor, PA, USA). The powder was sieved by a vibratory sieve shaker (AS 200, Retsch, Haan, Germany). For buffered powders, either borax or sodium carbonate was added to the mix before the freeze-drying step.

In the next step of the process, NU-phase powder was mixed with NHS-POx powder. Beforehand, NHS-POx powder was colored blue by the addition of 0.05% FD&C blue, a food-grade coloring substance. Acetone, here used as a desolvating agent, was added dropwise during pestle and mortar grinding to effectively form homogenized granules out of the mix. The granules were then dried for 24 h, and subsequently ground again by pestle and mortar and sieved by vibratory sieve shaker until a particle size of either 250–500 µm or 125–250 µm was reached. For powders containing a surfactant, Pluronic F-68 was either impregnated into NU-phase alongside NHS-POx or was coated on milled and sieved powder at 65 °C in a rotovap (Buchi, Hendrik-Ido-Ambacht, The Netherlands). The finished product was vacuum sealed and stored at −20 °C. The production process of all adhesive powders used in this study is displayed in Appendix A.

### 2.4. Chemical Analysis

Nuclear magnetic resonance (NMR) analysis (JEOL, Akishima, Japan) was performed to determine the NHS-ester degree of functionalization (NHS-DF) of NHS-POx in the powder [32]. NHS-DF is based on the relative peak intensity of NHS-ester at 2.8 ppm compared to a POx-sidechain reference peak at 4 ppm. The water content remaining in the powder was determined via Karl Fisher titration (Mettler-Toledo, Greifensee, Switzerland). The pH of the powder was determined using a pH meter and pH strips (VWR, Radnor, PA, USA). The detailed chemical characterization methods can be found in Appendix A. The specifications of all used powder prototypes in this study are found in Appendix A.

### 2.5. Wettability Tests

The wetting rates of powders, NU-phase, and individual ingredients (i.e., crosslinked gelatin powder, NHS-POx, NU-POx, trehalose) were determined by the Washburn capillary rise technique. We used a method similar to previous studies [44,45]. For this test, a sample (130 mg) was packed in a 3 mL syringe with 9.83 mm diameter (Henke-Sass Wolf, Tuttlingen, Germany) and taped off with grade 1 filter paper (VWR, Radnor, PA, USA). Here, the sample was pressurized with 500 g for 1 min to minimize pores between particles. Subsequently, the powder-filled syringe was slowly lowered towards a beaker filled with saline water (0.9% NaCl) that was standing on an analytical balance until the bottom of the tube touched the saline water. The liquid mass transfer from the beaker to the tube was recorded using a camera. The weights of the dry filter paper and the powder sample were documented before each test. The weight was documented a second before and all seconds until a minute after contact between saline and the tube, as well as the weight of the wet filter paper, were documented during each test. Wetting rates were extracted from the results by plotting the square of the mass of saline imbibed in the powder sample against time and calculating the slope of the second linear part of the graph, as described in earlier work [44]. The gathered data were also used to determine the time until a plateau in water uptake was reached, as well as the total water uptake per sample. After an equilibrium was reached, the (partially) gelled powder was removed from the cylinder, and a picture was taken to visualize the wetting of each sample.

### 2.6. SEM Imaging

For the room temperature scanning electron microscope (SEM), the powders were placed on SEM stubs with carbon tape. The samples were coated with chromium (Quorum Technologies, Laughton, UK) for conductivity and imaged with the Zeiss Crossbeam 550 FIB/SEM (Carl Zeiss Microscopy GmbH, Oberkochen, Germany). SEM images (SESI and InLens detectors) were collected at 2 kV acceleration potential and 200 pA probe current. For CryoSEM, powder hydrogels were formed by wetting the powders with milliQ water and incubating for 10 min. The resulting gel was cut and placed into a high-pressure freezing carrier (Ø 3 mm, depth 2 × 50 μm, Art. 390, Wohlwend) and frozen rapidly in 2100 atm in 2 msec using the high-pressure freezing machine µ-live (CryoCapCell, Paris, France). The frozen sample was then loaded into a cryo-holder and transferred to the Quorum preparation chamber (PP3010T, Quorum Technologies, Laughton, UK) for sublimation at −100 °C for 10 min and Platinum sputter coating at 5 mA for 45 s. After coating, the sample was transferred into the Zeiss Crossbeam 550 FIB/SEM (Carl Zeiss Microscopy GmbH, Oberkochen, Germany). Samples were imaged using both the SESI and InLens detectors using 2 kV acceleration potential and low (100 pA) probe current to avoid beam damage. Throughout imaging, the samples were kept at −160 °C and the anticontaminator at −185 °C. Two forms of the powder were tested to visualize differences in granulate and polymer matrix morphology.

### 2.7. Swelling Measurement

A volume-driven swelling test was performed based on earlier work [46,47]. In this experiment, 50 mg of powder per sample was added to a graduated cylinder. The cylinder was tapped to create a homogeneous powder layer, and the volume of the dry powder sample was noted. Subsequently, 5 mL of phosphate-buffered saline (PBS) was added to the sample, which was then incubated at 37 °C. Volumes were determined after 2 and 24 h of incubation, and swelling rates were determined by comparing the volume of the dry sample to the wetted sample at different time points (Equation (1)). Experiments were performed in triplicate.(1)$$Q=\frac{{Volume}_{maximum}}{{Volume}_{initial}}\times 100\%$$

### 2.8. Crosslinking Speed Measurement

Crosslinking speed of the powder was tested in an inverted vial test, as described in the literature [47]. During this test, 50 mg samples of powder were added to glass vials, and crosslinking was initiated by adding 200 µL of saline to the vial, to reach a concentration of 250 g/L. The tubes were then subsequently flipped on designated time points (1–10 s), until a gel had formed that stayed in place. The powder was tested in various forms to determine the effect of pH, hydrophilicity, and particle size on its gel-forming abilities.

### 2.9. In Vitro Degradation Measurements

Per sample, 250 mg of powder was crosslinked with 1 mL PBS until a gel was formed. The gel was then submerged in 40 mL PBS and incubated at 37 °C (n = 3) or 70 °C (control, n = 1). The mass of the gel, together with the solvent pH, was measured every 7 days for 6 weeks. The mass was measured by placing the gel on an analytical scale (VWR, Radnor, PA, USA), and the solvent pH was measured by using a pH strip.

Furthermore, solvent was extracted from all samples every 7 days for 6 weeks for NMR analysis. Here, we analyzed the solvent for the presence of polymer-bound NHS, free NHS ester, and succinic acid, where we determined the concentration of each compound using an internal standard. Internal standards were made for every compound by measuring the peak intensity of 3 different concentrations of the compound in triplicate. For the integral conversion of free NHS and succinic acid peaks, the detected concentrations beyond 0.5 mg/mL were considered reliable, which is the limit of detection of the standard curves. For the polymer-bounded NHS group, the detected concentrations beyond 0.35 mg/mL were considered reliable (Appendix A). At every time period, sufficient liquid was extracted from the samples to perform ^1^H-NMR analysis twice. This experiment was performed in triplicate.

### 2.10. Adhesive Strength Measurements

#### 2.10.1. Treatment and Characterization of Collagen Sheets

Collagen sheets (Nippi, Tokyo, Japan) were used as a substrate during adhesive strength measurements. The sheets were activated to increase the availability of amine groups on the surface, as was performed in earlier research [48]. Collagen sheets were incubated at 80 °C in phosphate-buffered saline (PBS, pH 7.4) and EDTA for 20 min. Thereafter, collagen sheets were placed in HEPES buffer and NaOH (pH 8.0) and were stored at refrigerated temperatures (2–7 °C). Collagen sheet activation was determined by testing the tensile strength of a certified ETHIZIA^®^ patch with a material testing machine (Zwick-Roell, Ulm, Germany) in triplicate per batch of prepared collagen sheets. No difference in tensile strength was observed between collagen sheet batches used in this study).

#### 2.10.2. Tensile Strength

Powder prototypes were compared to powder with deactivated NHS-POx (i.e., POx-OH-COOH, a POx polymer with hydrolyzed NHS-ester groups) as a negative control, and a fibrin glue as a positive control in a tensile strength test. In this test, based on ASTM F2258-05 [49], the tensile adhesion strength to tissue was measured by determining the force needed to linearly pull apart two pieces of two collagen sheets, adhered together by the powder or a control. The collagen sausage casings (2.5 × 5 cm^2^) were cut, and single sheets were fixed with superglue (Sika, Dublin, Ireland) to the 2.5 × 2.5 cm^2^ surfaces of two 3D-printed grip tabs. These grip tabs were specifically designed to minimize the effect of external forces on the adhesion during application, incubation, and transport of the sample to and from the incubator (Figure 2a). The application of all powders (NHS-POx- and POx-OH-COOH-containing powders) was performed as follows: A 200 mg powder sample was spread evenly onto the stationary grip tap, resulting in a thin and homogeneous layer to ensure optimal sample wetting. Thereafter, the mobile grip tab was placed on top, and both grip tabs were immediately placed in 37 °C saline water, with the top grip tab pressed on the stationary grip tab by 5 N for 1 min. After the pressure was removed, the sample was incubated in saline water (NaCl 0.9%) at 37 °C for 1 h. The application of fibrin glue was based on its instructions for use [50] and was performed as follows: a 0.6 mL sample of fibrin glue was used to create a thin, homogenous layer on the stationary grip tab without sample spillage. Then, the mobile grip tab was placed on top and pressed on the stationary grip tab by 5 N for 1 min, without any added saline water, after which the glue was hardened for 3 min before being placed at 37 °C in saline water for 1 h. Upon removal of the sample from the incubator, the grip tabs were connected to a mechanical bench tester and pulled apart at a speed of 5 mm/min. Maximum force, failure type, and sample wetness were determined during this experiment using TextXpert III (Version 1.91, Zwick-Roell, Ulm, Germany), which was performed with 6 repetitions per experimental group.

#### 2.10.3. Shear Strength

The shear strength of powder prototypes, as well as inactivated powder with POx-OH-COOH as a negative control and fibrin glue as a positive control, was examined in a shear strength test. The test was based on ASTM F2255-05 [51] to determine the specific shear strength by measuring the force needed to pull apart two pieces of collagen sheets in opposite directions, glued together by a powder prototype or a control group. Custom grip tabs were designed to promote stability during application and incubation, which had surfaces of 1 × 2.5 cm^2^ (Figure 2b). Collagen sheets (2.5 × 5 cm^2^) prewetted with saline were fixated to the grip tab surfaces using superglue, and the sheet on the stationary grip tab was patted dry using a gauze (Hartmann, Heidenheim, Germany). For these surfaces, 80 mg of powder and 0.3 mL fibrin glue were needed to create a thin, homogeneous layer. Here, the sample weight-to-surface area ratio was maintained with the tensile strength method to ensure comparability between adhesive strength tests. The procedure for product application, incubation, and strength testing was similar to the tensile strength test.

#### 2.10.4. Adhesion Durability

The durability of the wet adhesive strength of powder prototypes was tested by performing the tensile strength test and extending the incubation time from 1 h to 7 and 14 days. Here, powder was crosslinked with 5 N pressure in 37 °C saline for 60 s. Afterwards, the cup containing both the grip tab structure and the saline was sealed with parafilm before it was placed in the incubator to prevent any evaporation of the saline from occurring. This process was performed in an aseptic manner to prevent contamination. Powder prototypes and fibrin glue were tested in triplicate per time point in this experiment.

### 2.11. Statistical Analysis

Statistical analysis was performed on all wettability tests and all adhesive strength measures. Means and standard deviations were calculated using the MEAN and STDEV.P options in Microsoft Excel (Version 2507, Redmond, WA, USA) and by using the program R in RStudio (Version 2024.12.0, Posit, Boston, MA, USA). An Analysis of Variance (ANOVA) test was performed on means and standard deviations of specific subgroups in an experiment to determine the presence of a significant statistical difference between these subgroups. Thereafter, a Tukey’s HSD test was performed to determine the statistical significance between subgroups. Tukey’s test was chosen as we were interested in pair-wise comparisons between group means, and as the sample size for each group was equal. Subsequently, groups were categorized via the Compact Letter Display method, which was added to the figures of the plots. For all variables with the same letter in a figure, the difference between the groups is not statistically significant. Two groups are significantly different from one another when different letters are depicted [52].

## 3. Results

### 3.1. Chemical Characteristics of Ingredients, NU-Phase, and Powder

The powder appears light blue due to the mixture of blue-colored NHS-POx and the white NU-phase (Figure 3a). A robust gel formed upon the addition of water (Figure 3b), and the gel exhibited adhesive properties when crosslinked between two surfaces containing amine groups (Figure 3c).

NMR analysis of NHS-POx displayed the presence of a peak at 2.8 ppm, indicating the presence of NHS-ester groups in the polymer (Figure 4a). The NMR spectrum of the powder showed similar relative peak intensities to the NHS-POx spectrum, with peaks at 2.8 ppm (NHS-ester) and 4 ppm (reference), respectively (Figure 4b). This indicates that no loss of NHS-ester groups has occurred during the granulation of NU-phase and NHS-POx.

The wetting rate of trehalose dihydrate was 23,962 ± 11,746 mg^2^/s, which was significantly higher than all other ingredients (*p* < 0.01, Figure 5, Appendix A). NU-phase, which consisted for 60% *w*/*w* out of trehalose, had a wetting rate of 3168.9 ± 989 mg^2^/s, which was significantly lower than trehalose (*p* = 0.0041). This indicated a successful impregnation of trehalose (and NU-POx) into the crosslinked gelatin network, where the hygroscopic ability of trehalose is physically hindered.

### 3.2. SEM Imaging

SEM images of dry powder showcased the differences between particle sizes. Powder granulates with a larger particle size showed comparable structures (Figure 6a), while morphological differences are more clearly visible for smaller sized powder particles (Figure 6b). When the powder is crosslinked into a gel, a dense network is formed with fibrils reaching 1 µm and pore sizes of 50–500 nm (Figure 6c). A powder containing both a buffer and a surfactant forms a gel with fibrils of 400–600 nm, with pore sizes of 100–700 nm (Figure 6d).

### 3.3. Swelling Measurement

After 24 h, the adhesive powder increased to 112 ± 8% its dry volume, in comparison to a 114 ± 6% increase of NU-phase and a 386 ± 8% increase of untreated crosslinked gelatin powder (Figure 7a). The powder only increased in volume during crosslinking; no swelling was observed after the polymer matrix was already formed (Appendix A).

### 3.4. Crosslinking Speed Measurement

A powder with 125–250 µm particles formed a gel after 10 s that stayed in place after vial inversion (Figure 7b). A bigger particle size of 250–500 µm increased the time needed to form a sufficient hydrogel to 12 s. This could be caused by the fact that a powder with larger particle size has a lower surface area to volume ratio compared to a powder with a smaller particle size. Crosslinking speed was slightly higher when Pluronic F68 was added to the powder mix (9 s for 125–250 µm, 12 s for 250–500 µm). When a buffer (Borax) was used to increase the powder’s pH to 7.4, the crosslinking speed strongly increased (2–3 s for 125–250 µm and 250–500 µm).

### 3.5. In Vitro Degradation Measurement

The mass of the formed gel at the start of the experiment was 1.21 ± 0.0112 g for 37 °C samples and 1.24 g for the 70 °C sample (Figure 8a). The mass of gel incubated at 37 °C increased during the first 7 days (2.09 ± 0.153 g), before declining at 14 days (0.231 ± 0.327 g). After 14 days, two of the triplicate samples were completely dissolved in PBS, while one sample had a gel mass of 0.694 g. No gel mass was measurable after 21 days. Gel was completely dissolved in PBS after 1 day of incubation at 70 °C. The solvent pH of the gel incubated at 37 °C showed a negative trend over 6 weeks, in which the pH went from a neutral 7 to an acidic 5.5. Solvent pH of gel incubated at 70 °C reached 5.5 after 7 days, which then did not change for the rest of the measurement.

Free NHS ester was the most present in fluid samples out of all examined byproducts after 1 day incubation (Figure 8b). Here, concentrations decreased from 1.82 mg/mL to 0.5 mg/mL in the first 21 days of incubation at 37 °C, where the concentration stabilized in the days thereafter. Polymer-bound NHS ester concentration at 37 °C increased for the first 21 days until a concentration of 3.96 mg/mL was reached, after which a decrease in concentration was observed. The increasing presence of polymer-bound NHS ester in fluid samples in the first 21 days visualizes the degradation of the POx polymer matrix, until no visible gel structure was left. Succinic acid concentrations in 37 °C increased in concentration starting from 7 days of incubation, where a concentration of 1.65 mg/mL was reached after 42 days of incubation. This indicates the further breakdown of the POx polymers, where the previously NHS-containing sidechain increasingly breaks off from the polymer backbone. Analysis of individual samples displayed that one sample (tube 3) had a different degree of degradation than the other two, thereby causing a large deviation for polymer-bounded NHS concentration at 21 days and succinic acid concentration at 14–42 days (Appendix A). After 7 days of incubation at 70 °C, succinic acid was the only byproduct with a significant signal in the solvent sample, while free NHS and polymer-bound NHS concentrations were negligible after 7 days (Figure 8c).

### 3.6. Creation of Powder Prototypes

Based on the previously described results, powder prototypes **P1**–**P7** were created to examine the effects of increased pH or hydrophilicity on the powder’s water uptake and adhesive capabilities (Table 1, Appendix A). All powder prototypes were made in both smaller- and larger particle sizes (125–250 µm and 250–500 µm, respectively).

### 3.7. Wettability Measurement

Powder prototypes **P1**–**P7** had wetting rates between 3205 and 12,766 mg^2^/mL, which were higher than NU-Phase and most powder ingredients, but less than trehalose (Figure 5 and Figure 9). For smaller particle sizes, the wetting rates of powders with Pluronic F68, both impregnated (**P2** and **P4**) and coated (**P6** and **P7**), were significantly higher than for unbuffered powder (**P1**, *p* = 0.0048, *p* = 0.028, *p* = 0.0024, *p* = 0.014, respectively), although this was not found in Pluronic F68-impregnated powder buffered with carbonate (**P5**, *p* = 0.070). There were no differences in wetting rate, time to reach plateau, and total water uptake between the two tested methods of Pluronic F68 impregnation and coating (**P2** and **P4** vs. **P6** and **P7**, Figure 9 and Figure 10, Appendix A).

The wetting rate of powder with only borax added (**P3**) was not different from that of unbuffered powder (**P1**) for both large and small particle sizes (*p* = 0.45 and *p* = 0.43, respectively). Additionally, buffered powder did not reach a plateau earlier compared to non-buffered powders. A trend was present for powders with acidic pH, where bigger particle size powders took longer to reach a plateau and had a higher total water uptake than their smaller-sized form, even though a statistical difference was not consistently found (**P1**, **P2**, **P6**, Figure 10). This trend was not found for powders with a neutral pH.

Wetting rates between different particle sizes of the same powder prototype were comparable, with **P1** and **P5** the only prototypes displaying differences, though not statistically significant (*p* = 0.057 and *p* = 0.14, respectively). The smaller particle size variant of **P5** took longer to reach a plateau compared to the bigger particle size variant (*p* = 0.047), while no difference in total water uptake was found (*p* = 0.93).

### 3.8. Adhesive Strength Measurements

Powder prototypes showed an average tensile strength from 14.96 ± 2.908 kPa to 19.34 ± 2.503 kPa (Figure 11a). The tensile strength of powders containing POx-OH-COOH was significantly lower compared to all other groups (*p* < 0.001). The tensile strength of smaller-sized **P1**, **P2**, **P3**, **P6**, and **P7**, and larger-sized **P2** and **P5,** was significantly higher compared to fibrin glue (*p* < 0.05). No significant differences in tensile strength between the prototypes were found. Smaller-sized **P1** had the highest average tensile strength, which was unexpected as it had the lowest wetting rate and crosslinking speed of all prototypes.

The average shear strength of **P1** and **P4** was 30.63 ± 6.699 kPa and 27.14 ± 2.691 kPa, respectively, while for **P5**, shear strength averaged 15.01 ± 2.189 kPa (Figure 11b). Post-hoc analysis indicated a significantly higher shear strength of **P1**, **P4**, and **P5** compared to fibrin glue (*p* < 0.00001) and powder containing POx-OH-COOH (*p* < 0.0001). **P1** and **P4** displayed significantly higher shear strength than **P5** (*p* = 0.00000060 and *p* = 0.000034, respectively).

For adhesion durability, the tensile strength of **P1** (both particle sizes), **P2**, **P6,** and **P7** was significantly higher than fibrin glue after 7 days (*p* < 0.05, Figure 11c). The adhesive strength of buffered powder prototypes decreased more quickly than that of unbuffered powders, indicative of the powder’s pH being closer to the optimum pH for NHS hydrolyzation. The buffered powders **P3**, **P4**, **P5**, and **P7** had the lowest average tensile strength of all powders after 7 days. Buffered powder coated with Pluronic F68 (**P7**) portrayed a higher adhesive strength after 7 days of incubation than buffered powder without Pluronic addition (**P3**, *p* = 0.020).

After 14 days, the tensile strength of only **P3** and **P7** was significantly higher than fibrin glue (*p* = 0.0025 and *p* = 0.0015, respectively, Figure 11d). Smaller-sized **P1** had the highest average tensile strength (14.51 ± 8.919 kPa), even though no statistical significance was found compared to fibrin glue (*p* = 0.06). No difference in tensile strength was found between the Pluronic F68 addition methods in unbuffered prototypes (**P2** vs. **P6**, *p* = 0.95), while buffered powder with coated Pluronic F68 performed significantly better than buffered powder with Pluronic F68 impregnated (**P4** and **P5** vs. **P7**, *p* = 0.0019 and *p* = 0.0032, respectively).

## 4. Discussion

The aim of the current study was to characterize a novel POx-based tissue adhesive for obliterating surgical dead space. The swelling, crosslinking speed, and degradability properties of the powder led to the creation of various prototypes, different in pH, hydrophilicity, and particle size. These prototypes were subsequently tested for their water uptake and adhesive properties, two relevant features for clinical applicability and effectiveness.

Buffer added to the powder did not lead to a higher wetting rate or an enhanced adhesive strength. It was expected that introducing a buffer would improve the powder’s wetting rate and adhesive strength, as a higher pH increases NHS-ester reactivity, potentially leading to more extensive amide bond formation. Next to comparable wetting rates, buffered powders did not reach a plateau sooner compared to non-buffered powders and did not absorb more water during wettability tests. Images taken after tests showed that the buffered powder samples were comparably wet to the unbuffered powders (Appendix A). An alkaline buffer can lower a product’s surface tension and subsequently enhance water penetration, similar to the effects of added surfactants [53,54]. This initiates rapid gel formation, which may hinder further solvent penetration in the upper layers of the powder, leading to only partial wetting during wettability tests. It is possible that both mechanisms counterbalance each other, resulting in comparable wetting rates and adhesive strength compared to unbuffered powders, at least in in vitro conditions.

Powders with carbonate as a buffer agent displayed different results compared with those with borax as a buffer agent. Large variations in wetting rates were found in the larger-sized powder containing carbonate, but not in the smaller-sized powder, while the shear strength of powder containing carbonate was significantly lower in comparison with borax-containing powder. This may be attributed to the release of CO_2_ gases from carbonate upon contact with water [55], which could compromise the stability of the added buffer, reducing its usability.

Pluronic F68 addition, either by impregnation or coating, increased the wetting rate. An increase in wetting rate for Pluronic F68-containing powders was expected, as the addition of Pluronic F68 increases the powder’s hydrophilicity. Pluronic F68 should be more accessible to improve solvent uptake when coated on powder instead of impregnation, while Pluronic F68 impregnation should increase the solubility during crosslinking [56]. As we observed similar wetting rates for powders containing Pluronic F68, it seems that both methods of Pluronic addition have a comparable effect on the powder’s wetting rate.

The addition of both buffer and Pluronic F68 did not improve adhesive strength and led to a reduction in adhesive durability. Combining buffer and surfactant was expected to be beneficial for the powder, as the powder’s increased hydrophilicity could counteract the blockage of solvent movement during gel forming. However, Pluronic F68- and buffer-containing powders not only have a similar adhesive strength to other prototypes, but also the shortest adhesive durability. Both buffer- and Pluronic F68 addition lowers the surface tension, increasing water penetration [57,58]. Simultaneously, the more alkaline environment leads to higher hydroxide concentration, which actively breaks down amide bonds via nucleophilic attacks [59]. This creates an environment where nucleophilic attacks from hydroxide are more effective, resulting in a more rapid gel degradation. This mechanism is also found when biosurfactants are used to accelerate the biodegradation process of non-biodegradable polymers [60].

For powder with an acidic pH, bigger particle size powders had a higher total water uptake and took longer to reach a plateau than smaller particle size powders (**P1**, **P2**, **P6**, Figure 9). However, this distinction was not visible in powders with a neutral pH. Although these results seem to fall in line with what was found during crosslinking speed experiments, it could also be an artifact from the Washburn capillary model that was used. Larger particles can create larger pores when packed in comparison to smaller particle sizes, which can lead to irregular liquid penetration. Also, it is known that not all pores are filled at an even rate, which could lead to deviations in measurement [44,45]. The bed porosity was assumed to be consistent across samples of the same size, owing to the standardized sample packaging procedure employed in the Washburn method. We did not explicitly test for potential differences in porosity related to varying particle sizes.

The difference in adhesive strength between powder prototypes and fibrin glue was larger during shear strength testing than during tensile testing. This was mostly caused by the fact that the tensile strength of fibrin glue was strikingly higher than its shear strength [61]. Fibrin forms a three-dimensional network by the ligation of fibrin molecules to create strong fibrin branches in the longitudinal orientation, thereby increasing material stiffness. These branches are also ligated in lateral orientation, but this is mostly less pronounced than in the longitudinal direction, which results in significantly lower shear strength capabilities [62,63]. In POx-based powders, the amide bonds that constitute the polymer matrix are oriented in all directions, allowing them to effectively dissipate mechanical forces from all angles [33]. An adhesive that would obliterate dead space should be able to handle mechanical stress from all directions, which makes a POx-based adhesive more capable of handling variable stressors than fibrin glue.

The adhesive properties of POx-based tissue adhesives were mechanically more in tune with neighboring tissues than those of other tissue adhesives. The novel adhesive powder did not show volumetric swelling after the first hour and only increased to 1.12 times its size. This is in contrast to other adhesives used in clinics, especially hydrophilic polymer adhesives based on PEG, acrylic acid, collagen, thrombin, and cellulose [64,65,66,67,68,69]. Excessive swelling can put increasing pressure on neighboring tissues, which can have detrimental physiological effects [24,25]. Furthermore, excessive swelling could dilute crosslinking density, which can diminish adhesive strength [29,70,71]. The adhesive strength of the powder prototypes was comparable to subcutaneous tissue layers. The adhesive strength of subcutaneous fat was determined to be 4.77 kPa, as tested in an ex vivo rat model [72]. The biomechanical strength of fascia has been tested in previous studies, which showcase a variety of strength profiles between fascia tissue from different locations in the body [73]. The Young’s modulus of subcutaneous tissue has been determined to be between 15 and 50 kPa, as tested on subcutaneous tissue from different anatomic locations [74]. With 15–20 kPa as the maximum tensile force, the strength of the gels formed by POx-based adhesives is therefore within the range of subcutaneous fat and fascia tissue. Other commercially available tissue adhesives are significantly stiffer than internal tissues [30,75]. This mechanical mismatch can impair tissue movement, which increases patients’ discomfort, and can cause additional tissue damage, leading to a stronger inflammatory response and excessive scarring [29,76]. A stiffer wound area could alter wound healing pathways, which can diminish the proliferation and differentiation of cells needed for wound healing [77,78,79].

For the obliteration of surgical dead spaces, POx-based tissue adhesives benefit from their need for moisture from wet tissues to initiate crosslinking. Current tissue adhesives, such as cyanoacrylates and fibrin glues, require dry surfaces for effective adhesion, which significantly limits their applicability on wet surfaces, which are commonly present after surgical injury [11,29,80]. An example of this is the difference in the application technique between POx-based tissue adhesives and fibrin glue during the adhesive strength experiments as described in this study. For POx-based adhesives, the full grip tab structure was immersed in saline immediately upon applying pressure to both grip tabs. This simultaneous fixation and saline application are crucial for crosslinking initiation as saline dissolves NHS-POx, thereby allowing for the formation of covalent bonds between NHS-POx and amine-containing compounds. Furthermore, applying pressure minimizes the volume of the area in which the powder can form a matrix and reduces the distance between the tissue layers intended for adhesion. For fibrin glue, the application of fibrin glue was performed on a completely dry surface. After applying pressure for one minute, wetting with saline had to be avoided for three minutes at room temperature to allow the fibrin glue to stabilize, in accordance with the instructions for use [50]. While moisture, commonly present at the wound site, is detrimental to fibrin adhesives, the moisture extruding from tissue layers enhances the matrix formation of POx-based tissue adhesives. Nevertheless, it is important to note that we selected TISSEEL as one of the leading commercially available sealants, despite its intended use being different from obliterating dead spaces.

It is still unclear what the ideal adhesion durability and biodegradability of this tissue adhesive should be. Recovery time until a wound is healed in the average surgical patient is difficult to pinpoint. As a general rule of thumb, clinicians and researchers have described wound healing lasting shorter than 21 days to be “normal” and wound healing lasting more than 21 days to be “chronic” [81,82,83], although others argue that this dictum should be extended [84]. Drainage duration could also be indicative of wound closure, as drains are generally removed when drainage volume drops below a certain threshold, which is after 1–3 weeks postoperatively for patients undergoing breast surgery [6,8,85]. Based on this knowledge, the powder prototypes that still display adhesive strength after 14 days seem to have more preferable characteristics for clinical use. However, more research, especially in more clinically relevant models, is needed for confirmation of the most appropriate adhesive durability and biodegradation timelines.

### 4.1. Limitations

The substrate used for adhesive strengths experiments was specifically treated, which make the results of this study not directly comparable to adhesive strength results from other experimental studies. The amine density in the human body is different per type of tissue [86]. Nippi sausage casings are made with type I collagen [87], which is extracted from skin, tendons, vasculature, and bone, and which resembles the amine density of our tissues of interest (muscle, fat, and fascia) more than other sources of collagen. The collagen that is present in the collagen sheets has fewer amine groups available for covalent crosslinking than collagen in subcutaneous tissue, as most amine groups present on the collagen sheet are used during the sheet formation process [88]. To tackle this, a preparation method was used to activate sausage casings before use in adhesive strength measurements, as was previously performed by others [48]. In this method, secondary and tertiary structures of collagen strands are partially degraded, so that more amines become available for covalent binding [89].

Simulated body fluids could have been used for in vitro degradation and adhesive durability experiments. Durability and degradation measurements in the current study were performed in saline and PBS, respectively, both of which have been used often during in vitro experimentation of new medical devices. Some experimental studies have made use of solutions that resemble body fluids, such as wound exudate or lymphatic fluid [90], while others have added collagenases to the solvent during in vitro degradation experiments to mimic enzymatic degradation of the product [80,91]. The ionic composition of simulated body fluids is similar to human blood plasma, while the ionic composition of PBS is largely different [92]. Still, the improvement in translatability when using simulated body fluids for in vitro experiments is debatable, as the translational gap with in vivo degradational experiments remains large [11,93]. This makes in vivo survival studies the most important method to determine the true adhesive durability and degradational process of the POx-based adhesive powder, as has been shown with comparable products containing NHS-POx [34,94].

This study did not address safety, toxicity, or other biocompatibility test results. The current research primarily focused on the mechanical properties of the novel POx-based powder and the different prototypes. With further ex vivo (perfusion) organ and tissue experiments of animal and human tissue, we want to further select powder prototypes. Later phases of product development will involve in vivo studies to assess the safety and toxicity of a smaller group of prototypes. These in vivo experiments, particularly histological analyses, will provide comprehensive data on key cellular processes such as inflammation, necrosis, fibrosis, and cell proliferation and differentiation, ensuring thorough evaluation of the tissue adhesive’s biocompatibility. This study does not include any ex vivo or in vivo experimental data. The primary objective was to evaluate the fundamental properties of various POx-based powder prototypes, such as water uptake, crosslinking kinetics, and adhesion strength and durability. The in vitro characterization experiments employed are faster and more cost-effective; however, they offer limited translatability compared to ex vivo or in vivo studies. The most promising formulations will undergo further evaluation during ex vivo tests. Subsequently, a select subset of top-performing candidates will be assessed in in vivo studies. This phased approach efficiently screens out inferior prototypes through scalable in vitro experiments, thereby ensuring that only the most promising candidates progress to more costly and physiologically relevant tests. Importantly, this strategy minimizes animal suffering by reducing the number of lab animals exposed to suboptimal prototypes.

### 4.2. Future Directives

The results of the current study provide a basis for further evaluating powder prototypes on more elaborate models, which should be designed to resemble the clinical situation. 3D-oriented defect models using animal tissue would provide relevant information on the adhesive capabilities of the prototypes on irregular surfaces and relevant tissue types like fat and fascia tissue. Cadavers or perfused tissue models could be used for the creation of a subcutaneous cavity defect model, where the efficacy of the adhesive prototypes to obliterate subcutaneous dead spaces, as well as their ideal dosage and mode of action, could be evaluated before in vivo animal studies. In addition, mastectomy and/or axillary dissection models can be performed in cadavers to test the effect of powder dosage and distribution on adhesion efficacy. Finally, an animal model would be the ideal method to test the adhesive’s biocompatibility, degradability, and adhesive durability.

## 5. Conclusions

This study demonstrates the characteristics of a novel POx-based tissue adhesive that holds promise for obliterating subcutaneous dead spaces after surgery. The adhesive powder is characterized by limited swelling, rapid crosslinking, and robust adhesion. The crosslinking speed is accelerated when a basic buffer or a surfactant is added to the mix, while an increase in particle size slows down the crosslinking speed. Multiple powder prototypes displayed higher immediate adhesive strength in comparison to a commercially available fibrin glue. The adhesive durability is shortened when both a basic buffer and a surfactant are present. The impact of alterations in the formulation on adhesive capabilities underlines the potential for product tweaking dependent on the desired indication. The current study has provided sufficient evidence to discriminate between prototypes for further ex vivo and in vivo studies.

## Figures and Tables

**Figure 1 bioengineering-12-01011-f001:**
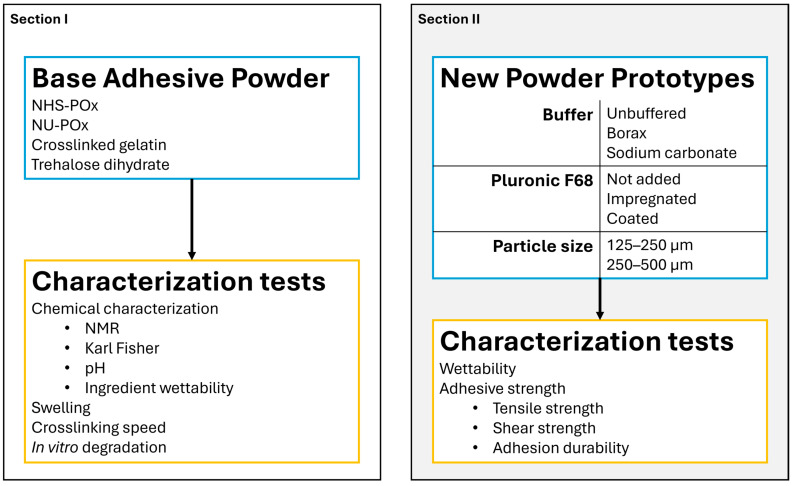
Schematic displaying the study design outline. The two panels describe the different sections of the study. Blue text boxes describe the details of the adhesive powder(s) used. Orange text boxes describe the characterization tests that were performed.

**Figure 2 bioengineering-12-01011-f002:**
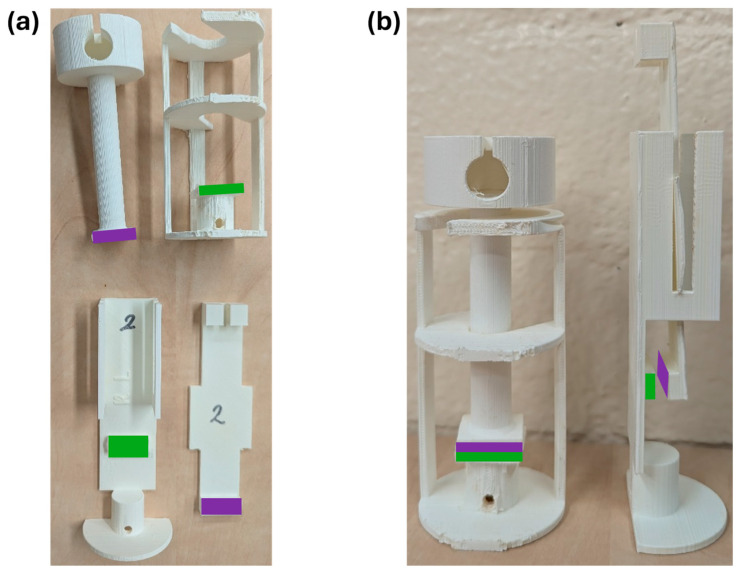
3D printed grip tabs used for adhesive strength measurements: (**a**) Separate grip tabs for tensile strength (**above**) and shear strength (**below**); (**b**) Grip tabs for tensile strength (**left**) and shear strength (**right**) in stacked position. Surface areas where collagen sheets were glued to are marked green and purple on the stationary and the mobile grip tab, respectively.

**Figure 3 bioengineering-12-01011-f003:**
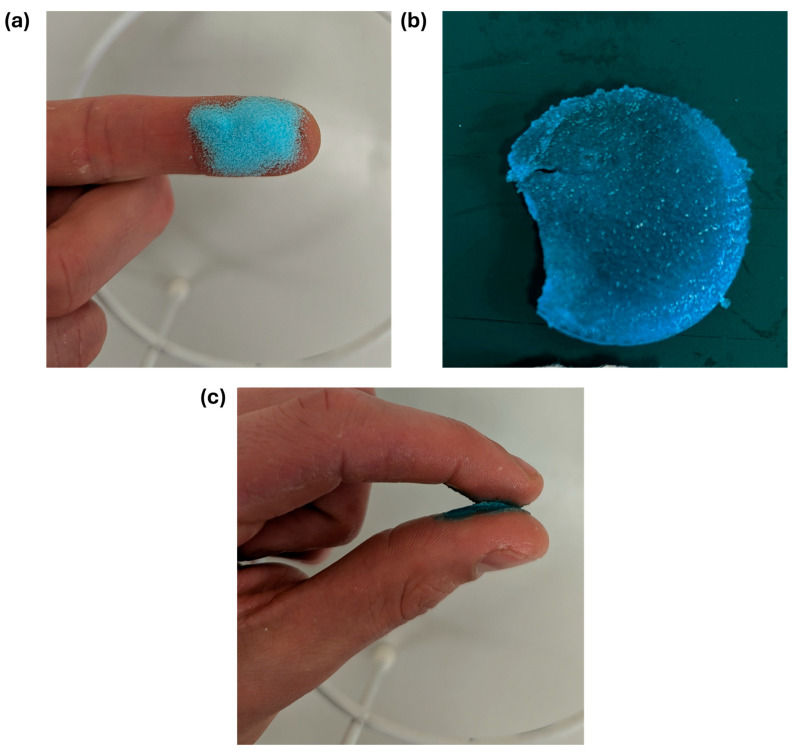
Photos of the adhesive powder are (**a**) dry powder, (**b**) crosslinked gel, and (**c**) crosslinked between amine-containing surfaces.

**Figure 4 bioengineering-12-01011-f004:**
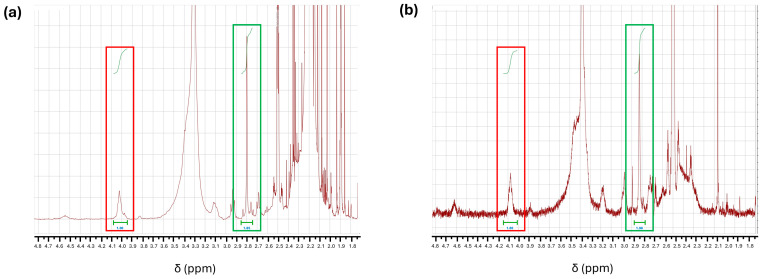
NMR spectra of: (**a**) NHS-POx and (**b**) adhesive powder containing NHS-POx. For both figures, peaks at 4 ppm and 2.8 ppm are indicated with a red and a green rectangle, respectively.

**Figure 5 bioengineering-12-01011-f005:**
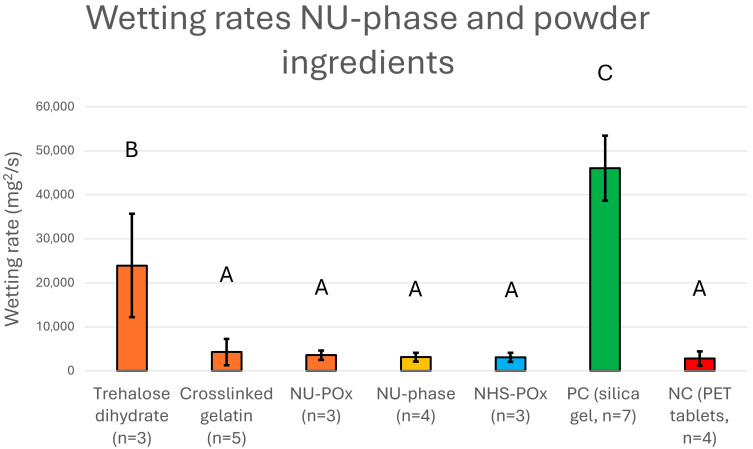
Wetting rates of individual powder ingredients and NU-phase in comparison to a positive control (PC) and a negative control (NC). Error bars are calculated from standard deviations. Letters (A, B, C) indicate the significant differences (*p* < 0.05) between subgroups as calculated via ANOVA and Tukey HSD test following the compact letter display method.

**Figure 6 bioengineering-12-01011-f006:**
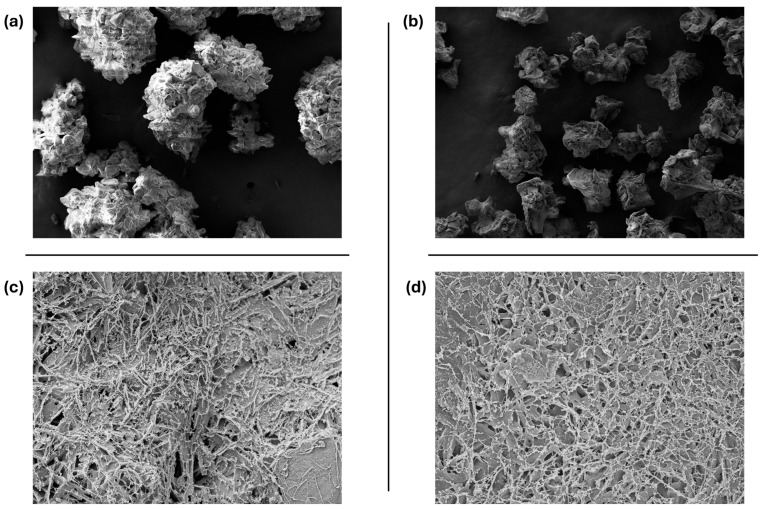
Room temperature SEM images of (**a**) dry powder particles of a powder of larger particle size (250–500 µm) and (**b**) of smaller size (125–250 µm). Room temperature SEM images are 100 times magnified. (**c**) CryoSEM images of the crosslinked gel structure of powder without buffer and surfactant, and (**d**) of a powder containing buffer and surfactant. CryoSEM images are 7000 times magnified.

**Figure 7 bioengineering-12-01011-f007:**
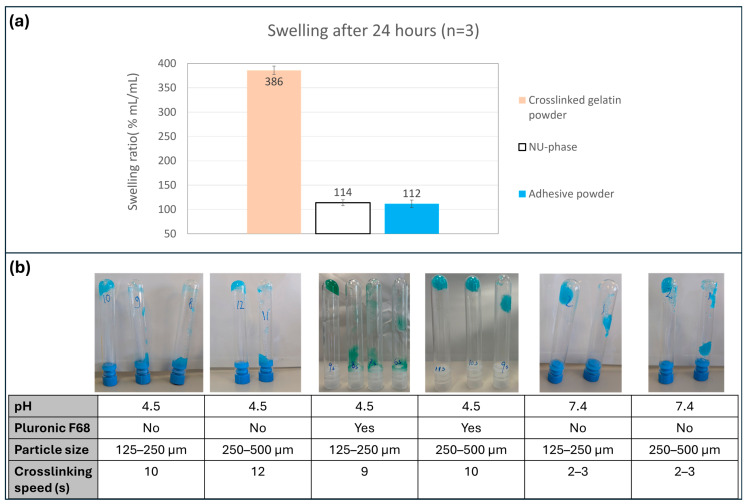
(**a**) Bar chart displaying volumetric swelling ratio of non-processed crosslinked gelatine powder, NU-phase, and adhesive powder. Bar indicates average swelling ratio; error bars indicate standard deviation; (**b**) Crosslinking speed of adhesive powder variants differing in pH, presence of Pluronic F68, and particle size. Images indicating the time needed for the powders to form a gel that stays in place when the tubes are inverted.

**Figure 8 bioengineering-12-01011-f008:**
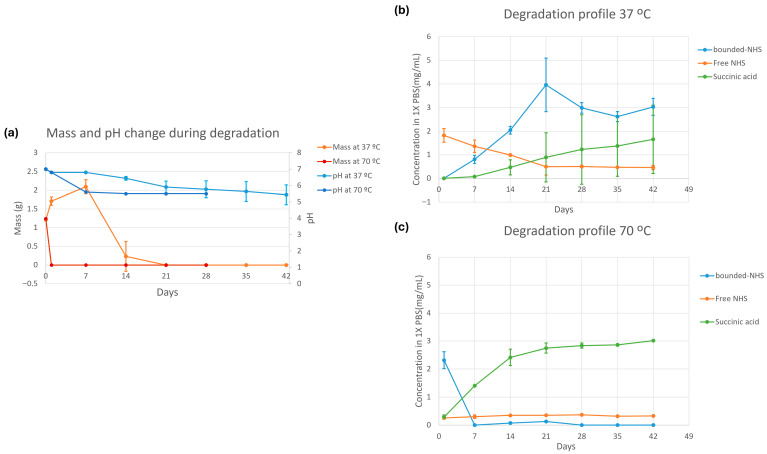
In vitro degradation of crosslinked adhesive powder in 1× PBS: (**a**) Mass and pH change of crosslinked adhesive powder at 37 °C and 70 °C; (**b**) Degradational products of crosslinked adhesive powder at 37 °C and (**c**) 70 °C. The sample solvent was extracted from the sample and analyzed with NMR. All experiments were performed in triplicate, and each sample was measured twice per time period. Error bars indicating standard deviations.

**Figure 9 bioengineering-12-01011-f009:**
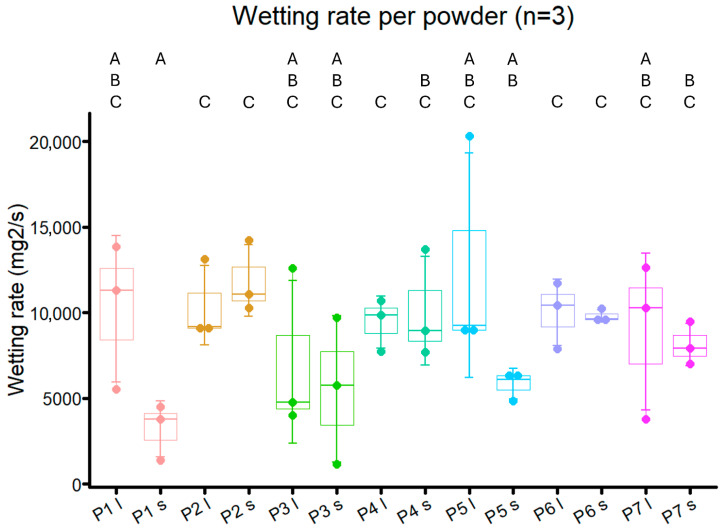
Wetting rate of powder prototypes **P1**–**P7**. l indicates a particle size of 250–500 µm, s indicates a particle size of 125–250 µm. A–C are used to indicate statistical differences between groups, with *p* < 0.05.

**Figure 10 bioengineering-12-01011-f010:**
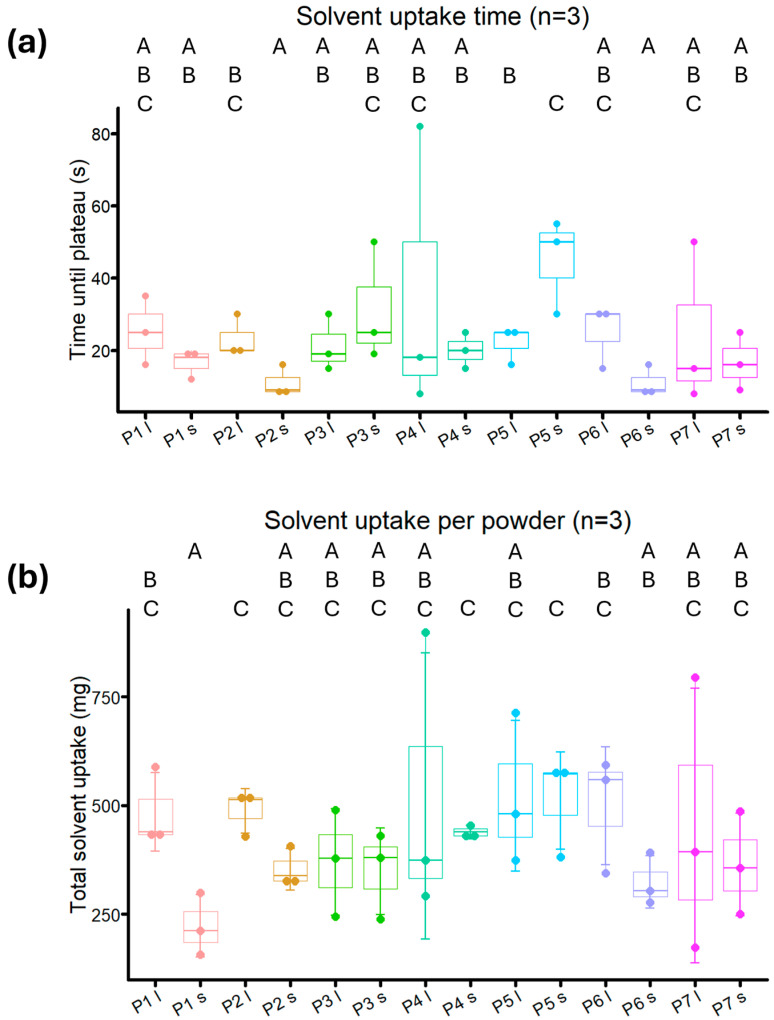
(**a**) Total experimental time until a plateau in solvent uptake was reached per sample, and (**b**) Total solvent uptake per sample, for powder prototypes **P1**–**P7**. l indicates a particle size of 250–500 µm, s indicates a particle size of 125–250 µm. A–C are used to indicate statistical differences between groups, with *p* < 0.05.

**Figure 11 bioengineering-12-01011-f011:**
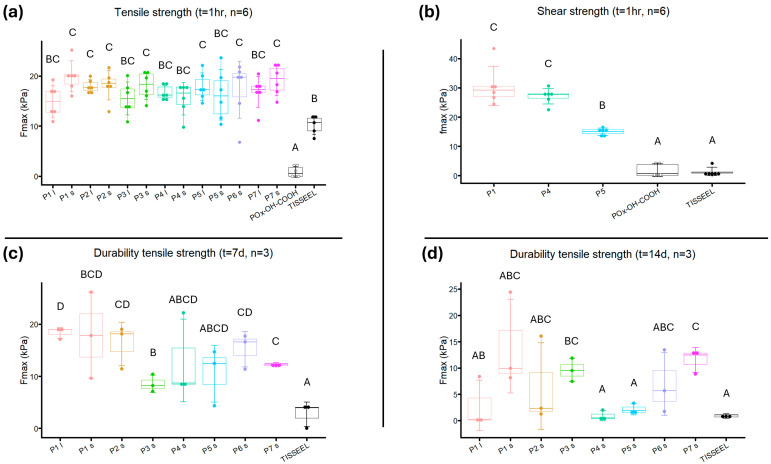
Dot plot of: (**a**) Tensile strength of adhesive powders with different particles sizes after 1 h incubation in saline water at 37 °C; (**b**) Shear strength of adhesive powder prototypes P1, P4 and P5 (all 125–250 µm) after 1 h incubation in saline water at 37 °C; (**c**) Tensile strength of prototypes after 7 day incubation in 37 °C saline water, and; (**d**) tensile strength of prototypes after 14 day incubation in 37 °C saline water. Negative control powder containing POx-OH-COOH is indicated in white, and positive control fibrin glue is indicated in black. Error bars indicate standard deviations higher and lower than the mean. A–D are used to indicate statistical differences between groups, with *p* < 0.05.

**Table 1 bioengineering-12-01011-t001:** Description of the powder prototypes used for wetting rate and adhesion testing.

Prototype	Buffer Added to Powder	Pluronic F68 Added to Powder
**P1**	No	No
**P2**	No	Impregnated
**P3**	Borax	No
**P4**	Borax	Impregnated
**P5**	Sodium carbonate	Impregnated
**P6**	No	Coated
**P7**	Borax	Coated

## Data Availability

The data that support the findings of this study are available from the corresponding author (S.E.M.P.) upon reasonable request.

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
