# Peer review of "Characterization of a Novel POx-Based Adhesive Powder for Obliterating Dead Spaces After Surgery"

_bioengineering, 2025, doi:10.3390/bioengineering12101011_

Round 1

Reviewer 1 Report

Comments and Suggestions for Authors

In the manuscript, the authors prepared a Characterization of a Novel POx-Based Adhesive Powder for Obliterating Dead Spaces After Surgery. And the provided manuscript is very well organized, but some minor changes in this manuscript. Thus, I recommend that this manuscript can be accepted for publication in bioengineering. The specific comments are as follows:
-English language and style aren’t well, and moderate English changes are required. Please give more information in the Introduction part about application part. 
-write full name of NHS-POX in the abstract.
- provide the representation of scheme.
-What is the novelty of this work?

Author Response

  • English language and style aren’t well, and moderate English changes are required. Please give more information in the Introduction part about application part. 

Answer: The application procedure of the adhesive powder has been added to the introduction to inform the viewer on the preferred usage of the powder.

Introduction, page 2:

“We have developed an adhesive powder based on NHS-POx intended to obliterate dead space and prevent hematoma and seroma after soft tissue and tissue flap surgery. When applied, the powder is sprayed on a dried wound site to form a thin layer. During wound closure, moist present enables covalent amide bonds to form between NHS-ester sites on NHS-POx and amine sites present in the powder and in tissue. These bonds help bridge the defect, effectively closing the dead space.”

  • write full name of NHS-POX in the abstract.

Answer: NHS-POx is now fully written out in de abstract.

Abstract, page 1:

“An adhesive powder based on N-Hydroxysuccinimide-poly(2-oxazoline), NHS-POx, combines robust adhesive strength in moist environments with favorable biocompatibility and biodegradability, which makes this an interesting candidate for eliminating spaces that remain between tissues after surgery.”

  • provide the representation of scheme.

Answer: A flowchart has been created which more clearly indicates the study outline. This flowchart is added as a figure, and a caption has been added that explains flowchart’s details.

Materials and Methods, Study outline, page 4:

Figure 1 Schematic displaying of the study design outline. The two panels describe the different sections of the study. Blue text boxes describe the details of the adhesive powder(s) used. Orange text boxes describe the characterization tests that were per-formed.”

  • What is the novelty of this work?

Answer: The product characterized in this study is an adhesive powder based on polyoxazoline (POx) polymers. While POx polymers have not yet been extensively utilized in medical applications, they exhibit significant potential due to their highly tuneable properties. The advantages of POx-molecules primarily stem from their structural versatility, as each individual monomer can be modified with various sidechains that influence the polymer’s core chemical characteristics, including solubility, hydrophilicity, stability, and crosslinking kinetics1. POx-polymers containing NHS-esters as sidechain (NHS-POx) has been optimized to rapidly form amide bounds with amine-containing compounds, rendering these polymers particularly suitable for use as tissue adhesives. Previous studies have demonstrated the successful application of NHS-POx in the development of a haemostatic patch2, a lung sealant3 and a bone-adhesive membrane4. In this study, we present a sprayable powder formulation based on NHS-POx technology, leveraging its adhesive properties to expand potential applications within the surgical setting.

More text on novelty has been added to the introduction and is also incorporated in the abstract:

Introduction, page 2

“Polyoxazoline polymers offer the flexibility to incorporate side chains on a per-monomer basis, rendering them highly tuneable for a wide range of specific applications.”

Reviewer 2 Report

Comments and Suggestions for Authors

This manuscript presents the development and characterization of a novel POx-based adhesive powder designed to address the clinical challenge of closing surgical dead spaces. The material demonstrates minimal swelling, rapid crosslinking, strong adhesion, and tunable physicochemical properties, with performance in several respects superior to fibrin glue. The work is innovative and addresses an important unmet clinical need, providing a strong foundation for further research. Nevertheless, there are several areas that require clarification or additional data to improve the rigor, interpretability, and translational relevance of the study.

Major Comments:

Mechanistic explanations:The observed effects of pH, surfactant addition, and particle size on crosslinking kinetics and adhesive durability are clearly described, but the mechanistic explanations are somewhat limited. For example, the proposed reasons for the reduced durability with combined buffer and Pluronic F68 (hydrolytic degradation from excess water molecules, potential hydrogen peroxide species) should be supported with additional discussion, literature, or experimental evidence.

Structural characterization: Incorporating SEM or other imaging techniques to analyze the morphology of crosslinked gels (e.g., pore size, network density) could provide valuable insights into how formulation parameters (e.g., surfactants, particle size) influence material performance, particularly swelling and adhesion.

Biocompatibility and safety considerations:While prior studies are cited, this work would be substantially strengthened by including cytotoxicity data for the newly developed prototypes. Even basic assays (e.g., cell viability, metabolic activity, proliferation) in relevant cell types such as fibroblasts or adipocytes would better support the claims of favorable biocompatibility. Without this, safety and translational potential remain speculative.

In vitro vs. in vivo relevance: The current degradation and durability assessments are performed in saline or PBS, which are useful first steps but do not fully mimic physiological environments. Testing in simulated body fluids or other biologically relevant media would provide a more realistic picture of how the adhesive behaves under in vivo conditions. The translational gap should be more explicitly addressed.

Durability and degradation timelines:The study reports degradation up to 6 weeks; however, optimal timeframes for biodegradation and adhesive strength in the context of wound healing are not yet clear. Future directions should discuss what durability profiles are clinically desirable for different surgical applications.

Minor Comments:

Figures: Ensure consistent formatting in figure captions, including statistical annotations (letters A, B, C). The reference to "Figure 2A/B" in the caption for Figure 1 appears to be a numbering error and should be corrected.

Clarity: Results and discussion could benefit from more detailed explanation of observed trends to aid the reader’s understanding.

Overall Assessment:This study is a promising and innovative contribution to the field of surgical adhesives. Addressing the points above particularly biocompatibility testing, deeper mechanistic discussion, and additional structural characterization would considerably strengthen the manuscript’s impact and translational relevance.

Comments on the Quality of English Language

The manuscript is generally understandable, but several sentences are long and complex, which reduces clarity. Shorter, more direct sentences are recommended.

Author Response

  • Mechanistic explanations: The observed effects of pH, surfactant addition, and particle size on crosslinking kinetics and adhesive durability are clearly described, but the mechanistic explanations are somewhat limited. For example, the proposed reasons for the reduced durability with combined buffer and Pluronic F68 (hydrolytic degradation from excess water molecules, potential hydrogen peroxide species) should be supported with additional discussion, literature, or experimental evidence.

Answer: We want to thank the reviewer for the proposed improved discussion.

The discussion of the differences between the adhesive powder prototypes mainly focused on expected versus observed results. The discussions on the findings were primarily descriptive and based on similar instances found in the literature. In our revision, we have expanded this perspective by incorporating sources from various disciplines that could offer valid explanations to our findings. This is how we improved the example that the reviewer has described above:

Discussion, page 21:

“The addition of both buffer and Pluronic F68 did not improve adhesive strength and even caused an earlier reduction in adhesive strength. Combining buffer and surfactant was expected to be beneficial for the product as the powders’ increased hydrophilicity could counteract the blockage of solvent movement during gel forming.  However, Pluronic F68- and buffer-containing powders did not only have a similar adhesive strength than other prototypes, but also the shortest adhesive durability. Both buffer- and Pluronic F68 addition lower the surface tension, increasing water penetration5, 6. Simultaneously, the more alkaline environment leads to higher hydroxide concentration, which actively breaks down amide bounds via nucleophilic attacks 7. This creates an environment where nucleophilic attacks from hydroxide are more effective, resulting in a more rapid gel degradation. This mechanism is also found when biosurfactants are used to accelerate the biodegradation process of non-biodegradable polymers 8.”

  • Structural characterization: Incorporating SEM or other imaging techniques to analyze the morphology of crosslinked gels (e.g., pore size, network density) could provide valuable insights into how formulation parameters (e.g., surfactants, particle size) influence material performance, particularly swelling and adhesion.

 Answer: We agree with the reviewer. We have now performed SEM analyses on multiple powder prototype samples. We used a powder with a particle size of 250-500 µm, and another prototype with 125-250 µm also containing Borax as buffer and Pluronic F-68 as surfactant. The SEM method that we used for the performance of this experiment has been added to the Methods section, while results have been added as Figure ... in the Results section.

Materials and Methods, page 5-6

“SEM imaging

For room temperature scanning electron microscope (SEM), The powders were placed on SEM stubs with a carbon tape. The samples where coated with chromium (Quorum technologies, Laughton, UK) for conductivity and imaged with the Zeiss Crossbeam 550 FIB/SEM (Carl Zeiss Microscopy GmbH, Oberkochen, Germany). SEM images (SESI and InLens detector) were collected at 2 kV acceleration potential and 200 pA probe current. For CryoSEM, powder hydrogels were formed by wetting the powders with milliQ water and incubating for 10 min. The resulting gel was cut and placed into a high pressure freezing carrier (Ø 3 mm, depth 2x50 μm, Art. 390, Wohlwend) and frozen rapidly in 2100 atm in 2 msec using the high-pressure freezing machine µ-live (CryoCapCell, Paris, France). The frozen sample was then loaded into a cryo-holder and transferred to the Quorum preparation chamber (PP3010T, Quorum technologies, Laughton, UK) for sublimation at -100 ºC for 10 minutes and Platinum sputter coating at 5 mA for 45 seconds. After coating, the sample was transferred into the Zeiss Crossbeam 550 FIB/SEM (Carl Zeiss Microscopy GmbH, Oberkochen, Germany). samples were imaged using both the SESI and InLens detector using 2 kV acceleration potential and low (100 pA) probe current to avoid beam damage. Throughout imaging, the samples were kept at −160 °C and the anti-contaminator at -185 °C. Two forms of the powder were tested to visualize differences in granulate and polymer matrix morphology.”

Results, page 9

SEM imaging

SEM images of dry powder showcased the differences between particle sizes. Powder granulates with a larger particle size showed comparable structures (Figure 6a), while morphological differences are more clearly visible for smaller sized powder particles (Figure 6b). When the powder is crosslinked into a gel, a dense network is formed with fibrils reaching 1 µm and pore sizes of 50-500 nm (Figure 6c). A powder containing both a buffer and a surfactant forms a gel with fibrils of 400-600 nm, with pore sizes of 100-700 nm (Figure 6d).”

  • Biocompatibility and safety considerations: While prior studies are cited, this work would be substantially strengthened by including cytotoxicity data for the newly developed prototypes. Even basic assays (e.g., cell viability, metabolic activity, proliferation) in relevant cell types such as fibroblasts or adipocytes would better support the claims of favourable biocompatibility. Without this, safety and translational potential remain speculative.

Answer: We agree with the reviewer that cellular studies for viability, metabolic activity or proliferation would add valuable information to our current data. For this paper the focus was on the mechanical characterization of the adhesive powder and its prototypes. The information gained from the experiments in the current study provides us a solid basis to make a further selection of powders. The new selection of powder prototypes will be used for more elaborate ex vivo and in vivo experimentation, in which cytotoxicity and biocompatibility will be addressed.

As was mentioned by the reviewer, all ingredients used for the creation of the adhesive powder have been deemed biocompatible as based on decades of research and clinical evaluation. Crosslinked gelatin has been implemented in multiple medical devices over the past decades because of its favourable biocompatibility properties9. Trehalose is a non-reducing sugar widely used in both medicine and medical devices due to its biocompatibility, and its non-toxicity trait has been thoroughly examined 10. Pluronic, sodium carbonate and borax are also safe to use, and have been used as additions to medical devices in numerous occasions 11-15. Poly(2-oxazolines) have been described in literature as an enhancement of biocompatibility 1, 16. Both NHS-POx and NU-POx have been tested in preclinical and clinical tests, where no safety risks were found 17, 18.

We have now added the following extra text regarding biocompatibility and safety tests to the limitation paragraph of the manuscript.

Discussion, page 26:

“This study did not address safety, toxicity, or other biocompatibility test results. The current research primarily focused on the mechanical properties of the novel POx-based powder and the different prototypes. With further ex-vivo (perfusion) organ and tissue experiments of animal and human tissue we want to select powder prototypes. Later phases of product development will involve in vivo studies to assess safety and toxicity of a smaller group of prototypes. These in vivo experiments, particularly histological analyses, will provide comprehensive data on key cellular processes such as inflammation, necro-sis, fibrosis, and cell proliferation and differentiation, ensuring thorough evaluation of the tissue adhesive’s biocompatibility.”

  • In vitro vs. in vivo relevance: The current degradation and durability assessments are performed in saline or PBS, which are useful first steps but do not fully mimic physiological environments. Testing in simulated body fluids or other biologically relevant media would provide a more realistic picture of how the adhesive behaves under in vivo conditions. The translational gap should be more explicitly addressed.

Answer: The topic of simulated body fluids has been touched upon in the limitation part of the discussion. As was mentioned there, simulated body fluids could provide us with a more clinically relevant degradational process, although the significance of this improvement remains debatable. Currently, we are evaluating the product using a perfused tissue model, similar to methodologies described in previous studies 19-21. This model allows for the addition of body fluids, thereby enhancing its physiological relevance and the translational potential of the obtained data. Still, the most relevant way to determine the biodegradation of the tissue adhesive is to test the product in an in vivo experiment. Such a study will be performed in the future with a smaller selection of powder prototypes, as we have mentioned in the recommendations for future studies.

The following adjustment was made to the manuscript:

Discussion, page 23-24:

“Simulated body fluids could have been used for in vitro degradation and adhesive durability experiments. Durability and degradation measurements in the current study were performed in saline and PBS, respectively, both of which have been used often during in vitro experimentation of new medical devices. Some experimental studies have made use of solutions that resemble body fluids like wound exudate or lymphatic fluid22, while others have added collagenases to the solvent during in vitro degradation experiments to mimic enzymatic degradation of the product 23, 24. The ionic composition of simulated body fluids are similar to human blood plasma, while the ionic composition of PBS is largely different 25. Still, the improvement in translatability when using simulated body fluids for in vitro experiments is debatable, as the translational gap with in vivo degradational experiments remains large 26, 27. This makes in vivo survival studies the most important method to determine the true adhesive durability and degradational process of the POx-based adhesive powder, as has been shown with comparable products containing NHS-POx3, 28.”

  • Durability and degradation timelines: The study reports degradation up to 6 weeks; however, optimal timeframes for biodegradation and adhesive strength in the context of wound healing are not yet clear. Future directions should discuss what durability profiles are clinically desirable for different surgical applications.

Answer: The topic of durability and degradation is discussed in the last paragraph of the discussion before limitations. One of our most important take-aways of the study is the finding that different powder prototypes display different degradational profiles, which makes the powder tuneable in its biodegradation. The tuneability of the powder can be used to precisely adapt the product for ideal wound healing. The timeline of wound healing per tissue or per defect location remains difficult to pinpoint, although literature provides us with a classification of 2 weeks for first degree wound healing. On first hand, this info makes the powders that display adhesive strength after 14 days incubation the most promising candidates. However, it is yet to be determined what the effect the presence of the polymer network would be on wound healing processes. A future in vivo study would provide necessary answers to these questions, which is also what we recommend for future studies. This has been described in our manuscript:

Discussion, Future directives, page 24:

“The results of the current study provide a basis for further evaluating powder prototypes on more elaborate models, which should be designed to resemble the clinical situation. 3D-oriented defect models using animal tissue would provide relevant information on the adhesive capabilities of the prototypes on irregular surfaces and relevant tissue types like fat and fascia tissue. Cadaver- or perfused tissue models could be used for the creation of a subcutaneous cavity defect model, where the efficacy of the adhesive prototypes to obliterate subcutaneous dead spaces, as well as their ideal dosage and mode of action could be evaluated before in vivo animal studies. In addition, mastectomy and/or axillary dissection models can be performed in cadavers to test the effect of powder dosage and -distribution on adhesion efficacy. Finally, an animal model would be the ideal method to test the adhesive’s biocompatibility, degradability and adhesive durability and further select prototypes.”

Minor Comments:

  • Figures: Ensure consistent formatting in figure captions, including statistical annotations (letters A, B, C). The reference to "Figure 2A/B" in the caption for Figure 1 appears to be a numbering error and should be corrected.

Answer: All figure captions have been improved to conform to the layout style of Bioengineering. The mentioned numbering error has now been improved. As another figure has been added to the manuscript placed above the previous numbering error, it was not needed to adjust the numbering in the text.

Materials and Methods, Adhesive strength measurements, Tensile strength, page 6:

“The collagen sausage casings (2,5x5cm) were cut, and single sheets were fixated with superglue (Sika, Dublin, Ireland) to the 2,5x2,5 cm surfaces of two 3D-printed grip tabs. These grip tabs were specifically designed to minimize the effect of external forces on the adhesion during application, incubation and transport of the sample to and from the incubator (Figure 2a).”

  • Clarity: Results and discussion could benefit from more detailed explanation of observed trends to aid the reader’s understanding.

Answer: The overall language and clarity of the results and the discussion have been reevaluated to improve the reader’s experience.

Overall Assessment: This study is a promising and innovative contribution to the field of surgical adhesives. Addressing the points above particularly biocompatibility testing, deeper mechanistic discussion, and additional structural characterization would considerably strengthen the manuscript’s impact and translational relevance.

Reviewer 3 Report

Comments and Suggestions for Authors

Suggestions and Comments for the Authors:

  1. Primary Concern: The manuscript would benefit significantly from the inclusion of in vivo or ex vivo data to substantiate the findings. Comprehensive experimental evidence is essential to support the proposed application and enhance the manuscript’s impact. While the decision to include these data in this or a future manuscript rests with your group, I strongly recommend their inclusion here.
  2. Replace the generic keyword “powder” with a more specific term, such as “POx-based powder.”
  3. Include a flowchart illustrating the study design to improve clarity and transparency of the experimental process.
  4. Specify the version of any software used in the study.
  5. Consider merging Sections 3.7 and 3.8 into a single section discussing limitations and future directions. A deeper discussion would strengthen the manuscript.
  6. In the conclusion, emphasize the key findings to reinforce the manuscript’s contributions.

Author Response

  • Primary Concern: The manuscript would benefit significantly from the inclusion of in vivo or ex vivo data to substantiate the findings. Comprehensive experimental evidence is essential to support the proposed application and enhance the manuscript’s impact. While the decision to include these data in this or a future manuscript rests with your group, I strongly recommend their inclusion here.

Answer:

We agree with the reviewer that ex vivo or in vivo data would strengthen the current data on the novel adhesive powder. The current study describes the early developmental phase of the novel powder, where we evaluated multiple prototypes on their basic properties, such as water uptake and adhesion, through in vitro experiments. These experiments are faster and more cost-effective but less translatable than ex vivo or in vivo studies.

Our approach involves selecting the best-performing prototypes based on this study’s results for subsequent ex vivo testing. From these, we will identify the top candidates for in vivo proof-of-concept studies. This stepwise funnel strategy is our most practical option, as it quickly rules out weaker prototypes through scalable experiments. Only the top performers proceed to costly, highly translatable tests, minimizing animal suffering by avoiding unnecessary testing on lab animals of inferior prototypes.

Given this developmental stage, we prioritized establishing a solid foundational dataset before proceeding with more complex, translational experiments, which are deemed outside the scope of the current work.

Limitations

Discussion, Limitations, page 26

“This study does not include any ex vivo or in vivo experimental data. The primary objective was to evaluate the fundamental properties of various POx-based powder prototypes, such as water uptake, crosslinking kinetics, and adhesion strength and durability. The in vitro characterization experiments employed are faster and more cost-effective; however, they offer limited translatability compared to ex vivo or in vivo studies. The most promising formulations will undergo further evaluation during ex vivo tests. Subsequently, a select subset of top-performing candidates will be assessed in in vivo studies. This phased approach efficiently screens out inferior prototypes through scalable in vitro experiments, thereby ensuring that only the most promising candidates progress to more costly and physiologically relevant tests. Importantly, this strategy minimizes animal suffering by reducing the number of lab animals exposed to suboptimal prototypes.”

  • Replace the generic keyword “powder” with a more specific term, such as “POx-based powder.”

Answer: The keyword “powder” has been replaced by “POx-based powder”

Keywords, page 1:

“POx-based powder”

  • Include a flowchart illustrating the study design to improve clarity and transparency of the experimental process.

Answer: A flowchart has been created which more clearly indicates the study outline. This flowchart is added as a figure, and a caption has been added that explains flowchart’s details.

Materials and Methods, Study outline, page 4:

Figure 1 Schematic displaying of the study design outline. The two panels describe the different sections of the study. Blue text boxes describe the details of the adhesive powder(s) used. Orange text boxes describe the characterization tests that were per-formed.”

  • Specify the version of any software used in the study.

Answer: Software version of all used software was added in the Methods section.

Materials and Methods, Adhesive strength measurement, Tensile strength, page 7

“Maximum force, failure type, and sample wetness were determined during this experiment using TextXpert III (Version 1.91, Zwick-Roell, Ulm, Germany), which was performed with 6 repetitions per experimental group.”

Materials and Methods, Statistical analysis, page 8-9

“. Means and standard deviations were calculated using the MEAN and STDEV.P options in Microsoft Excel (Version 2507, Redmond, WA, USA) and by using the program R in RStudio (Version 2024.12.0, Posit, Boston, MA, USA).”

  • Consider merging Sections 3.7 and 3.8 into a single section discussing limitations and future directions. A deeper discussion would strengthen the manuscript.

Answer: The limitations section has been extended as requested by other reviewers. This has deepened the discussion on the experimental possibilities regarding the adhesive’s biodegradability and biocompatibility. The limitations section puts forward that in vivo experiments provide results with the highest quality for the examination of both the tissue adhesive’s biodegradability and biocompatibility properties during preclinical research. This type of research is recommended for future research, as mentioned in the future directives section. The improved alignment of the limitations- and future directives section adds value to the discussion that this manuscript wants to present. We believe it has merit that both sections remain in the revised version of the manuscript with the additions performed in each section.

  • In the conclusion, emphasize the key findings to reinforce the manuscript’s contributions.

Answer: Key findings of this study have been added to the conclusion.

Conclusions, page 26

“This study demonstrates the characteristics of a novel POx-based tissue adhesive that holds promise for obliterating subcutaneous dead spaces after surgery. The adhesive powder is characterized by limited swelling, rapid crosslinking and robust adhesion. The crosslinking speed is accelerated when a basic buffer or a surfactant is added to the mix, while an increase in particle size slows down the crosslinking speed. Multiple powder prototypes displayed higher immediate adhesive strength in comparison to a commercially available fibrin glue. The adhesive durability is shortened when both a basic buffer and a surfactant are present. This impact of alterations in the formulation on adhesive capabilities underlines the potential for product tweaking dependent on the desired indication. The current study has provided sufficient evidence to discriminate between prototypes for further ex vivo and in vivo studies.”

Reviewer 4 Report

Comments and Suggestions for Authors

        This paper reports a novel poly(2-oxazoline) (POx)-based NHS-activated powder adhesive designed to obliterate surgical dead spaces. The authors characterize wettability of the powder, adhesion on collagen, and long-term stability versus fibrin glue. the results show rapid wetting, strong adhesion, and partial durability, though buffer with Pluronic formulations lost strength by 14 days, highlighting a translational potential and optimization needs.

The paper can be published in this journal after some minor revisions. The comments are:

  1. How did author decide on the amounts of POx powder vs amounts of fibrin glue per cm² for the comparison? Did they consider normalizing to manufacturer-recommended dosing (e.g., per cm²) to better reflect clinical usage?
  2. Page 4, lines 161–169 “sample (130 mg) … 3 mL syringe taped off with filter paper … pressurized with 500 g for 1 minute … from slope of the second linear part …”
    Specify filter paper grade or brand, syringe inner diameter, and whether bed porosity and structure factors were measured or assumed (important for Washburn-based rates).
  3. Authors are suggested to provide a higher quality images of NMR spectra at figure 4, with larger font sizes of ppm and integrations values.
  4. In Figure 5, only the last two columns are bordered while the others are not. The figure should be reformatted to ensure consistency making the graph more professional and suitable for publication quality.

Author Response

This paper reports a novel poly(2-oxazoline) (POx)-based NHS-activated powder adhesive designed to obliterate surgical dead spaces. The authors characterize wettability of the powder, adhesion on collagen, and long-term stability versus fibrin glue. the results show rapid wetting, strong adhesion, and partial durability, though buffer with Pluronic formulations lost strength by 14 days, highlighting a translational potential and optimization needs.

The paper can be published in this journal after some minor revisions. The comments are:

  • How did author decide on the amounts of POx powder vs amounts of fibrin glue per cm² for the comparison? Did they consider normalizing to manufacturer-recommended dosing (e.g., per cm²) to better reflect clinical usage?

Answer: The amount of POx powder for tensile strength tests has been determined based on results from pilot tests. During these pilot experiments, it was determined that using a 200 mg powder sample produces a sufficiently thin and homogeneous layer, which facilitates optimal wetting during incubation. Higher sample masses resulted in less favorable wetting characteristics, potentially compromising the reliability and reproducibility of the adhesive strength measurements. The powder quantity for shear strength tests was extrapolated based on the tensile strength tests. To ensure comparability of adhesive strength outcomes between the two test methods, the weight-to-surface area ratio was maintained consistently across all experiments. TISSEEL volume was based on volume determinations as mentioned in its Instructions For Use29. We used 0.6 mL per sample, as a higher volume would lead to spillage, which would also comprise the reliability of the end results.

The following text was added to the Methods section:

Materials and Methods, Adhesive strength measurements, Tensile strength, page 7

“Powder prototypes were compared to powder with deactivated NHS-POx (i.e. POx-OH-COOH, a POx polymer with hydrolyzed NHS-ester groups) as negative control, and a fibrin glue as positive control in a tensile strength test. In this test, based on ASTM F2258-0530, the tensile adhesion strength to tissue was measured by determining the force needed to linearly pull apart two pieces of two collagen sheets, adhered together by the powder or a control. The collagen sausage casings (2,5x5cm) were cut, and single sheets were fixated with superglue (Sika, Dublin, Ireland) to the 2,5x2,5 cm surfaces of two 3D-printed grip tabs. These grip tabs were, specifically designed to minimize the effect of external forces on the adhesion during application, incubation and transport of the sample to and from the incubator (Figure 2a). The application of all powders (NHS-POx- and POx-OH-COOH-containing powders) were performed as follows: A 200 mg powder sample was spread evenly onto the stationary grip tap, resulting in a thin and homogeneous layer to ensure optimal sample wetting. Thereafter, the mobile grip tab was placed on top, and both grip tabs were immediately placed in 37 °C saline water, with the top grip tab pressed on the stationary grip tab by 5 N for 1 minute. After the pressure was removed, the sample was incubated in saline water (NaCl 0,9%) at 37 °C for 1 hour. The application of fibrin glue was based on its Instructions For Use29 and was performed as follows: a 0.6 mL sample of fibrin glue was used to create a thin, homogenous layer on the stationary grip tab without sample spillage.”

Materials and Methods, Adhesive strength measurements, Shear strength, page 8

“Here, the sample weight-to-surface area ratio was maintained with the tensile strength method to ensure comparability between adhesive strength tests.”

  • Page 4, lines 161–169 “sample (130 mg) … 3 mL syringe taped off with filter paper … pressurized with 500 g for 1 minute … from slope of the second linear part …”
    Specify filter paper grade or brand, syringe inner diameter, and whether bed porosity and structure factors were measured or assumed (important for Washburn-based rates).

Answer: Information on the syringe diameter and grade of filter paper have been added to the Method section:

Materials and Methods, page 24

“For this test, a sample (130 mg) was packed in a 3 ml syringe with 9.83 mm diameter (Henke-Sass Wolf, Tuttlingen, Germany) and taped off with grade 1 filter paper (VWR, Radnor, PA, USA).”

The bed porosity was assumed to be consistent across samples of the same size, owing to the standardized sample packaging procedure employed in the Washburn method. However, potential differences in porosity related to varying particle sizes were not directly measured. This info is added to the Discussion:

Discussion, page 24

“The bed porosity was assumed to be consistent across samples of the same size, owing to the standardized sample packaging procedure employed in the Washburn method. We did not explicitly test for potential differences in porosity related to varying particle sizes.”

  • Authors are suggested to provide a higher quality images of NMR spectra at figure 4, with larger font sizes of ppm and integrations values.

Answer: The NMR spectra of figure 4 have been adjusted to improve the clarity of the location of the integration values in the graphs. Red and green rectangles are added to indicate the peaks at 4 ppm and 2.8 ppm respectively.

Results, page 11

Figure 4. NMR spectra of: (a) NHS-POx and (b) adhesive powder containing NHS-POx. For both figures, peaks at 4 ppm and 2,8 ppm are indicated with a red- and a green rectangle, respectively.”

  • In Figure 5, only the last two columns are bordered while the others are not. The figure should be reformatted to ensure consistency making the graph more professional and suitable for publication quality

Answer: Figure 5 has been reformatted to improve consistency between bars.

Round 2

Reviewer 1 Report

Comments and Suggestions for Authors

The paper is acceptable in its current form.

Reviewer 2 Report

Comments and Suggestions for Authors

Accept as it is

Reviewer 3 Report

Comments and Suggestions for Authors

Although the primary concern was not fully addressed, I acknowledge the authors' rationale. The manuscript has undergone substantial improvement, and the authors have provided a thoughtful and satisfactory response. Therefore, I consider the manuscript suitable for publication.

Reviewer 4 Report

Comments and Suggestions for Authors

 The authors have addressed all the comments, and the paper is now suitable for publication in this journal.